# Comparison of Crosslinking Kinetics of UV-Transparent Ethylene-Vinyl Acetate Copolymer and Polyolefin Elastomer Encapsulants

**DOI:** 10.3390/polym14071441

**Published:** 2022-04-01

**Authors:** Gernot M. Wallner, Baloji Adothu, Robert Pugstaller, Francis R. Costa, Sudhanshu Mallick

**Affiliations:** 1Institute of Polymeric Materials and Testing & Christian Doppler Laboratory for Superimposed Mechanical-Environmental Ageing of Polymeric Hybrid Laminates (CDL-AgePol), University of Linz, Altenbergerstraße 69, 4040 Linz, Austria; robert.pugstaller@jku.at; 2Dubai Electricity and Water Authority (DEWA) Research & Development Center, MBR Solar Park, Dubai 564, United Arab Emirates; baloji.adothu@dewa.gov.ae; 3Borealis Polyolefine GmbH, St.-Peterstraße 25, 4021 Linz, Austria; francis.costa@borealisgroup.com; 4The National Centre for Photovoltaic Research and Education (NCPRE) and Metallurgical Engineering and Materials Science, Indian Institute of Technology Bombay, Mumbai 400076, India; mallick@iitb.ac.in

**Keywords:** EVA, POE, crosslinking kinetics, dynamic mechanical analysis, activation energy, photovoltaics

## Abstract

Encapsulants based on ethylene-vinyl acetate copolymers (EVA) or polyolefin elastomers (POE) are essential for glass or photovoltaic module laminates. To improve their multi-functional property profile and their durability, the encapsulants are frequently peroxide crosslinked. The crosslinking kinetics are affected by the macromolecular structure and the formulation with stabilizers such as phenolic antioxidants, hindered amine light stabilizers or aromatic ultraviolet (UV) absorbers. The main objective of this study was to implement temperature-rise and isothermal dynamic mechanical analysis (DMA) approaches in torsional mode and to assess and compare the crosslinking kinetics of novel UV-transparent encapsulants based on EVA and POE. The gelation time was evaluated from the crossover of the storage and loss shear modulus. While the investigated EVA and POE encapsulants revealed quite similar activation energy values of 155 kJ/moles, the storage modulus and complex viscosity in the rubbery state were significantly higher for EVA. Moreover, the gelation of the polar EVA grade was about four times faster than for the less polar POE encapsulant. Accordingly, the curing reaction of POE was retarded up to a factor of 1.6 to achieve a progress of crosslinking of 95%. Hence, distinct differences in the crosslinking kinetics of the UV-transparent EVA and POE grades were ascertained, which is highly relevant for the lamination of modules.

## 1. Introduction

Crucial components of glass laminates or photovoltaic modules are film adhesives, which are usually based on polar CHO macromolecules such as ethylene-vinyl acetate copolymer (EVA) or poly vinyl butyral (PVB) [1,2,3,4,5]. In recent years, less polar encapsulants based on polyolefin elastomers (POE) have been established [6,7,8,9,10]. While PVB requires an autoclave lamination process, EVA or POE are converted by the more time-efficient vacuum lamination.

Both EVA and POE require organic peroxide crosslinking to attain stable, robust and durable glass or PV module laminates [11]. During the lamination process, the macromolecular structure of the encapsulant changes from the non-crosslinked, entangled thermoplastic state into a widely meshed, three-dimensional network structure. After crosslinking, the encapsulant exhibits better thermal and UV stability, less mechanical creep, less degree of crystallinity and enhanced adherence to glass substrates, silicon solar cells, gridlines or busbars. In the curing process, vinyl silane adhesion promoters are covalently bonded to the macromolecular structure of the encapsulant and the silicate moieties of the glass [12]. A lower degree of the crystallinity of the encapsulants allows for an enhanced optical clarity [13]. However, improperly crosslinked encapsulants impose an adverse effect on the PV module performance. A common problem is the inhomogeneous distribution of the peroxide in the encapsulant films associated with the lateral variation of the crosslink density and an excess of a non-reacted corrosive peroxide [14,15,16]. As described in [17,18], some additives such as phenolic, nitroxyl and phosphite antioxidants lower the concentration of the peroxide-induced macroradical intermediates that support the polyolefin modifications. In contrast, UV-transparent additives based on hindered amine stabilizers (HAS) confer oxidative stability to the crosslinked EVA or POE materials without compromising the yields of the peroxide-initiated crosslinking [17,18]. Hence, the UV-transparent EVA or POE adhesives and encapsulants based on hindered amine stabilizers would allow for a more reliable lamination process.

To determine the crosslinking state, several methods have been established [19,20,21,22,23,24,25,26,27,28]. Soxhlet extraction (the ratio between the mass of the encapsulant film sample after and before extraction) is quite common in the industry and is standardized. However, it is time consuming and requires some material. Differential scanning calorimetry (DSC) and dynamic mechanical analysis (DMA) are much faster and more reliable methods to assess the curing kinetics and the progress of crosslinking [22,23]. In DSC experiments, a sample mass in the mg range is probed. The exothermic crosslinking reaction enthalpy is evaluated. This enthalpy value is affected by the environment (air vs. oxygen vs. nitrogen) [19,21,26]. The reaction enthalpy is amounted to a few J/g, especially in air or oxygen, and the experimental uncertainty is quite high [19].

In contrast, DMA in the molten state is much more sensitive to assess crosslinking kinetics. The real part of the viscosity and modulus undergoes a significant change of a few magnitudes. Moreover, a more representative sample with dimensions in the mm or g range is required for DMA. While temperature-rise DMA is performed at a fixed frequency, isothermal rheometry is based on time or frequency sweeps at a constant temperature. Non-isothermal temperature-rise tests are quite common to characterize thermal transitions, such as glass transitions, melting, the onset of the peroxide decomposition or the gelation of the encapsulant [19,20,21,22,23,24,26]. Such experiments allow for the assessment of the material changes that occur as they heat up in the lamination process. In contrast, isothermal rheometry better reflects the structural changes at the lamination temperature, which is commonly around 150 °C for peroxide-crosslinking encapsulants. Hence, isothermal DMA is quite relevant for mimicking the main process step of lamination. Interestingly, little research has been performed to assess the curing kinetics of EVA by isothermal DMA [20,22,24,26]. The final lamination quality and degree of crosslinking strongly depend on the peroxide concentration, additive formulation, lamination time, and temperature [23,26]. So far, no specific attention has been given to characterize the crosslinking kinetics of the UV-transparent EVA- or POE-based encapsulants.

Hence, the main objectives of this paper were to implement an isothermal DMA testing method for the assessment of the peroxide-initiated crosslinking kinetics of the encapsulants and to investigate and compare, for the first time, novel UV-transparent EVA and POE film adhesives, which were modified with hindered amine light stabilizers.

## 2. Materials and Methods

### 2.1. Encapsulant Materials and Films

Two UV-transparent, fast-cure, commercially available encapsulant films were investigated and compared: an ethylene-vinyl acetate copolymer film (EVA, F406PS^®^ from Hangzhou FIRST Applied Materials, Hangzhou, China), and a polyolefin elastomer film (POE, TF4 also from Hangzhou FIRST Applied Materials, Hangzhou, China). The thickness of the EVA and POE film was 0.45 and 0.54 mm, respectively. According to the data sheets, a gel content of more than 75% for EVA and 60% for POE was stated by the supplier. A qualitative stabilizer analysis of EVA and POE was performed using high-pressure liquid chromatography with UV and mass-spectroscopy detection [29,30,31]. The EVA and POE films were stabilized with Irgafos 126 (phosphite-based processing antioxidants) and Tinuvin 770 (a hindered amine light stabilizer (HALS)) with secondary amino groups (–NH–), ester ((C=O)–O–) linkage groups and an aliphatic (–C_8_H_16_–) central group.

The encapsulant films were stored in aluminum envelopes and kept in a vacuum box prior to characterization. The surface topology was assessed by laser confocal microscopy. While the investigated EVA film revealed a non-periodic surface topology with a maximum height difference of 130 µm, the pyramid-like surface topology of POE was periodic with a maximum height difference of 500 µm and a diagonal of 1.7 mm of the quadratic base of the pyramid.

### 2.2. Infrared Spectroscopy

To assess and confirm the chemical structure of the supplied encapsulant films, Fourier Transform Infrared (FTIR) spectrophotometry was performed in direct transmission mode using a PerkinElmer Spectrum 100 (PerkinElmer, Waltham, MA, USA). FTIR spectra were recorded in the range from 650 to 4000 cm^−1^ with 16 scans at a resolution of 4 cm^−1^.

### 2.3. Dynamic Mechanical Analysis (DMA)

DMA allows for the investigation of the viscoelastic properties of solids, gels or melts as a function of temperature, time or frequency at a given temperature. The changes in the viscoelastic properties of the EVA and POE encapsulants were measured using a Modular Compact Rheometer (MCR-502, Anton Paar, Graz, Austria). DMA was conducted in torsional mode from 20 to 200 °C at a frequency of 1 Hz at 0.1% strain. Isothermal DMA experiments were performed in torsional mode from 125 °C to 155 °C in 5 °C intervals at a frequency of 1 Hz. The shear stress was kept constant at 7000 Pa.

After 15 min of pre-heating of the oven at a constant temperature, the disc-shaped encapsulant film specimens were placed between the parallel plates with a diameter of 25 mm. The whole process of opening and closing the oven, placing the specimens, and starting the test took 10 s. Viscoelastic properties such as the storage modulus, loss modulus, and complex viscosity were recorded during the dynamic and isothermal tests. The gelation time (*t_gel_*) was obtained from the crossover point of the real and imaginary part of the shear modulus. By modelling the gelation time (*t_gel_*) using an Arrhenius approach, the activation energy values were deduced for both encapsulants. The complex viscosity data were evaluated as to the progress of the crosslinking reaction.

## 3. Results and Discussion

In the following, first the results of the IR spectroscopic investigations are described and discussed. Special attention was given to the POE encapsulant, which allowed for the qualitative assessment of the CHO(Si, N)-based comonomers, curing aids, stabilizers and adhesion promoters. In contrast, the EVA-specific peaks of the additives were partly overlaid by the strong absorptions of the vinyl acetate comonomer. In the second and third subchapter, as elucidated by the temperature-rise and isothermal rheological experiments, the similarities and differences of the crosslinking kinetics are described and discussed for the investigated EVA and POE films.

### 3.1. Structural Features of the Investigated Encapsulants

FTIR absorption spectra of the investigated EVA and POE films in the non-crosslinked reference (ref) and the fully cured (X) state are illustrated in Figure 1. The spectra were measured in transmission mode. Due to a film thickness of about 0.5 mm, the main peaks of the copolymer backbone (i.e., the resonant state of CH_2_ and CH_3_ stretching vibrations at 2920 and 2850 cm^−1^, of C=O stretching in EVA at 1730 cm^−1^, of CH_2_ bending vibrations at 1460 cm^−1^ or of ester-specific peaks in EVA at 1240, 1160 and 1020 cm^−1^) were already totally absorbing and could not be resolved. These specific peaks were confirmed and ascertained by the FTIR measurement in ATR mode. For a polar CHO comonomer content of about 10 m%, the C–O related vibrational peaks in the polar ethylene copolymers were totally absorbed in the transmission mode at the film thicknesses above 100 µm [32,33]. In contrast, the polar comonomer content of the investigated 0.5 mm thick EVA film was even higher (~32 m%), which confirmed the totally absorbing peaks in the carbonyl and ester absorption ranges of EVA.

For the investigated EVA film, absorbance peaks at the 1720, 1240, and 1020 cm^−1^ bands, which are also typical for ester and ether groups of cyanurate crosslinking additives, linkage groups of hindered amine light stabilizers or vinyl silane-based adhesion promoters [1,34,35], were totally absorbing. Hence, it was not possible for the investigated EVA film to deduce unambiguous information as to the additives from these peaks. Moreover, in the evaluation of the transmission spectra, special attention was given to the qualitative assessment of the comonomers, crosslinking agents, stabilizers and adhesion promoters in POE. For the POE encapsulant, pronounced peaks at 1790, 1720 and 1090 cm^−1^ were clearly discernible in the transmission spectra. The absorptions are presumably related to the C=O (at 1795, 1720 or 1090 cm^−1^) or Si–O (at 1090 cm^−1^) groups of the fast-cure, crosslinking agent tertiary butylperoxy-2-ethylhexylcarbonate (strongest peak at 1790 cm^−1^), the co-curing agent triallyl isocyanurate (tallest peak at 1700 cm^−1^), the hindered amine light stabilizer Tinuvin 770 (i.e., Bis(2,2,6,6-tetramethyl-4-piperidyl), sebacate (strongest peak at 1720 cm^−1^) or the Si–O group of the adhesion promoter 3-(trimethoxysilyl)propyl methacrylate (tallest peak at 1075 cm^−1^) [34]. Nevertheless, it should be mentioned that the pronounced peaks at 1720 and 1090 cm^−1^ could presumably also be related to low amounts (<10 m%) of the butyl acrylate comonomer in the investigated POE encapsulant [8]. In contrast to the C–H bonds, the C=O and Si–O groups were characterized by high integrated infrared absorption intensity values of more than 10,000 darks [36]. For a given film thickness of 0.5 mm, the measured absorption values, except for the peaks at 1720 and 1090 cm^−1^, resulted in the content of the additives being around 1 m%. According to [1,11,35], the EVA encapsulants are formulated with 0.1 m% of the HALSs and 1.5 m% of the peroxide curing agents. Due to the superposition of the additives related to the C=O and Si–O peaks with the totally absorbing carbonyl and ester peaks of the vinyl acetate comonomer, just small shoulders were discernible in the EVA spectra.

Peaks which were decreasing significantly upon crosslinking were marked and highlighted by the arrows. In agreement with the data provided in the literature for the peroxide-crosslinking agents or the curing reactions of polyolefins [37,38,39,40,41,42,43], the distinct absorption bands for POE or shoulders for EVA at 1790, 1765, 1410 or 1220 cm^−1^ were attributable to the fast-cure, crosslinking agent tertiary butylperoxy-2-ethylhexylcarbonate. Moreover, peaks at 1650 (for EVA), 990 (for both), 930 (for POE) and 810 (for EVA) cm^−1^ were detected in the non-crosslinked reference state. These bands, which were not discernable or just weakly absorbing in the fully cured state, are presumably related to the unsaturated C=C bonds of the co-curing agents (e.g., triallyl isocyanurate) or vinyl silanes. Hence, the investigated EVA and POE films were based on slightly differing curing agent formulations.

### 3.2. Temperature-Dependent Storage and Loss Modulus and Loss Factor

While the storage modulus (M′) is a measure for the elastic or reversible behavior, the loss modulus (M″) describes the viscous response of the encapsulant material. The shear modulus or viscosity of the encapsulant increases as the crosslinking reaction proceeds [23,24]. The thermal transitions of the peroxide-initiated crosslinking reaction such as the decomposition onset (T_on_), gelation point (T_gel_) and offset of the crosslinking reaction (T_off_) temperatures are displayed in Figure 2 and summarized in Table 1. In the temperatures ranging below 120 °C, the decay of the storage and loss modulus could be attributed to the enhanced inner mobility and melting of the crystal lamellae. The melting peak temperatures of EVA and POE were about 55 and 80 °C, respectively [8,19]. The onset of the crosslinking reaction associated with the minimum of the storage modulus was obtained at 125 °C for EVA and at 135 °C for POE. The onset of the crosslinking of EVA was in a similar range from 110 to 125 °C as compared to findings in the literature for the standard and fast-cure EVA encapsulants [8,19,23]. However, in this study, a novel UV-transparent EVA grade with HALSs was used. In contrast, in the literature, focus was given to UV-absorbing EVA grades with phenolic radical scavengers and aromatic UV stabilizers. As well described in [17,18], the peroxide decomposition and crosslinking reaction depends on both the chemical structure of the polymer and the stabilization package.

Due to the curing, an interpenetrating crosslink network is formed that is associated with a crossover of M′ and M″. This crossover point is termed the “gelation point” [24]. The gelation temperature and time were indicated by T_gel_ and *t_gel_*, respectively. A gelation temperature of 130 °C and 140 °C was obtained for EVA and POE, respectively. After gelation, the difference between M′ and M′′ became much more pronounced (Figure 2). A lower gelation temperature might induce more pronounced stresses on solar cells or ribbons during the lamination process. For the investigated EVA grade, a significantly lower flow capability was ascertained in the temperature range from 130 to 150 °C. At the commonly applied lamination temperature of 150 °C, a more than four times higher storage modulus value was deduced for the UV-transparent EVA encapsulant.

Both EVA and POE showed comparable offset temperature values (T_off_), slightly above 160 °C. Similar values in the range from 155 to 160 °C were reported in the literature for the EVA encapsulants [8,19,23]. At the offset temperature, M′ levelled off due to the completion of the crosslinking reaction. The following slight decrease indicated that there was still a non-crosslinked fraction. The supplier stated a gel content of more than 75% for EVA and 60% for POE. In agreement, the significantly higher storage modulus of the investigated EVA grade at a temperature of more than 165 °C might be an indication for an enhanced crosslinking density of EVA. Furthermore, this conclusion is confirmed by the significant differences of the loss factor values during the peroxide crosslinking. In the rubbery state, EVA revealed a loss factor of about 0.01, whereas it was close to 0.03 for POE.

Interestingly, the absolute, temperature-dependent storage modulus values of the investigated EVA and POE encapsulants differed from the data given in the literature [8,19,23]. While slightly higher values were obtained prior to the onset of the peroxide decomposition and curing, the storage modulus of EVA was lower in the cured state. Most likely, these differences can be attributed to variations in the measurement setup. While the experiments in [8,19,23] were run on a dynamic mechanical analyzer in tensile-shear mode using circular specimens of 9 mm in diameter, in this study, a plate-plate rheometer and well-defined torsional loading was employed. Moreover, more representative disc specimens with a diameter of 25 mm were characterized. For the investigated POE grade, the storage modulus values were markedly different from the data reported in [8]. Especially in the cured state, a factor of five lower M′ values were deduced in this study. Most likely, a quite dissimilar POE grade was investigated in [8]. The POE grades for the PV encapsulation are still under development and are therefore less standardized than the commercially available EVA encapsulants.

In the cured state, a factor of four higher M′ values were ascertained for the investigated EVA as compared to the POE grade. Considering a heating rate of 3 K/min, a total crosslinking reaction time of 13 and 9 min was deduced for EVA and POE, respectively. Hence, the curing of POE was taking place in a narrower processing window (temperature and time), while it was wider and characterized by a lower onset temperature for UV-transparent, fast-cure EVA.

### 3.3. Curing Kinetics, Activation Energy and Progress of Crosslinking

Isothermal storage and loss modulus curves are depicted in Figure 3 for EVA and POE cured at different temperatures ranging from 125 to 150 °C in 5 °C steps. For both encapsulants, the storage modulus was much more affected by the crosslinking than the loss modulus. Interestingly, EVA revealed a more pronounced change in the storage and loss modulus than POE. As long as the storage modulus is lower than the loss modulus (M′ < M″), the material is in the molten sol state. Upon onset of the crosslinking reaction, the storage modulus increases and crosses the loss modulus curve. This crossover point (M′ = M″) is called gelation and was used to evaluate the gelation time (*t_gel_*). At the gelation point, a three-dimensional, weakly crosslinked gel is established. There are still linear and non-crosslinked polymer chains available in the encapsulant. The network formation increases as the crosslinking reaction proceeds. Above the gelation point, a solid crosslinked state is achieved, resulting in the leveling off of the storage modulus. This indicates the completion of the peroxide crosslinking reaction. While the lower and upper bound of the storage modulus were almost independent of the test temperature, the gelation time was significantly lower at the elevated temperatures.

Comparison of the storage and loss modulus data of the investigated UV-transparent, fast-cure EVA with the values reported in [20,24] for a UV-absorbing EVA grade revealed a good agreement in the thermoplastic, non-crosslinked state. In the cured state, a factor of about 1.5 lower values were obtained in this study. Schulze et al. (2010 and 2015) also used a plate-plate rheometer and performed experiments in controlled-stress mode. The diameter of the plates and the specimens was 20 mm, in contrast to 25 mm in this study.

In Figure 3, the gelation points are indicated with open circles. The deduced gelation times at various isothermal curing temperatures are summarized in Table 2. In agreement with the temperature-rise experiments, a significantly faster gelation was ascertained for the investigated UV-transparent EVA encapsulant. Slightly dependent on the testing temperature, the gelation time was a factor of four times higher for the examined UV-transparent POE grade. A potential reason for the differences in the crosslinking kinetics is the dependency of the reaction rate on the primary structure of EVA and POE. Furthermore, there might be differences in the formulation with the peroxides and the co-curing agents. While the amount of the comonomer content for EVA is well described in the literature [1] and ranges from 28 to 33 w%, no details are given for the POE encapsulants. However, as mentioned in [8], the comonomer content of POE is significantly lower than for EVA, which would result in a lower concentration of tertiary carbon atoms along the main chain, and hence less crosslinking reactivity. In contrast to the standard cure EVA grade modified with UV absorbers and investigated in [22,26], the gelation time at 140 °C was about one magnitude faster for the UV-transparent, fast-cure EVA encapsulant in this study. A comparison with the data provided in [20,24] for another fast-cure, UV-absorbing EVA encapsulant showed a deviation of about 20% in the gelation time at 140 °C (60 vs. 71 s), but a comparable value of about 220 s at 130 °C.

To model the dependency of the gelation time on temperature, an Arrhenius fit was used. According to Arrhenius, the temperature-dependent kinetics of chemical reactions for a reaction rate (*k*) can be written as:(1)$$k\left(T\right)=A\;exp\left(-\frac{E_{a}}{RT}\right)$$
where *T* is the temperature, *A* is a material constant, *R* is the gas constant and *E_a_* is the activation energy. For many practical engineering implementations, the reaction rate *k* is insufficient to predict the activation energy. The gelation time or the time to achieve a specific threshold value is of higher interest. Hence, the Arrhenius equation can be re-written as:(2)$$lnt_{gel}=C-\left(\frac{E_{a}}{RT}\right)$$
where *C* is a specific material constant. Arrhenius variables alter when there is a change in the reaction mechanisms or a degradation of the polymeric network. It should be noted that the estimated reaction threshold time *t_gel_* is only valid in a specific temperature range.

Using Equation (2), an Arrhenius plot was deduced (see Figure 4). The activation energy (*E_a_*) was calculated from the slope of a linear fit of the Arrhenius plot. *E_a_* values of 155 and 154 kJ/moles were obtained for the investigated UV-transparent EVA and POE grades, respectively. These values were found at the 0.99 goodness of the linear fitting coefficient (R^2^). The comparable activation energy values are presumably related to the similar peroxide-curing and cyanurate co-curing agents added to the encapsulant formulations. The value obtained for the UV-transparent, fast-cure EVA grade of this study was significantly higher than the activation energy value of 124 kJ/moles reported in [20,24] for a UV-absorbing, fast-cure EVA encapsulant. While the gelation time for the investigated UV-transparent EVA grade was comparable at 130 °C, it was slower at 140 °C. Due to the unknown details of both of the commercially available EVA formulations, the ascertained differences could not be unambiguously attributed to the interactions of the curing and co-curing additives and aromatic UV absorbers or antioxidants. Nevertheless, as clearly evidenced in the literature, additives and stabilizers could have synergistic or antagonistic effects on peroxide-initiated crosslinking kinetics [17,18] and on long-term durability [30,31].

To assess and describe the progress of the curing reaction as a function of time, the complex viscosity (*η**) was evaluated. In Figure 5, the complex viscosity of EVA and POE is plotted for different temperatures. While the increase in the complex viscosity is related to the curing reaction, the leveling off indicates the completion of the crosslinking. The shape of the complex viscosity curves was almost similar for the EVA and POE encapsulants. However, a factor of more than four higher complex viscosity values were obtained for EVA in the cured state. This is in agreement with the results of the temperature-rise DMA. Presumably, the higher viscosity in the cured state of EVA could be attributed to the higher amount of tertiary carbon atoms, the denser crosslinking structure and the higher gel content. The initial complex viscosity values were at a comparable level. The rate of decomposition of the peroxides and the curing reactions were much faster at higher temperatures. Hence, the shear modulus and complex viscosity curves were shifted to shorter curing times at higher temperatures. A comparison of the complex viscosity data of the investigated UV-transparent EVA grade with values reported in [20,24] again revealed a good agreement in the non-crosslinked state and slightly higher values for the cured elastomer. Presumably, the attainable gel content was lower for the UV-transparent EVA grade.

The progress of the crosslinking reaction (*X*) (see Figure 6 and Table 2) was calculated from *η** by using Equation (3):(3)$$X=\frac{\eta^{*}\left(t\right)-\eta^{*}\left(t_{o}\right)}{\eta^{*}\left(t_{f}\right)-\eta^{*}\left(t_{o}\right)}$$
where *η**(*t*) is the complex viscosity at time *t*, *η**(*t_o_*) is the complex viscosity at the beginning of the experiment time (*t_o_*) and *η**(*t_f_*) is the final complex viscosity after reaching the saturation or levelled-off state.

The progress of the crosslinking reaction achieved at a specific gelation time was quite different for EVA and POE. For POE, a factor of more than six higher values of the progress of the crosslinking reaction values were deduced at the gelation time. Again, it was clearly confirmed that the crosslinking kinetics were significantly retarded for the investigated POE grade.

The required time to reach the recommended value of 95% of the progress of the crosslinking reaction [24] is summarized in Table 2. This control level of the crosslinking was achieved dependent on temperature by a factor of 1.2 to 1.6 times faster for the investigated EVA encapsulant. This retardation, which was more pronounced at higher temperatures, was not reflected unambiguously by the total time for the curing reaction obtained by the temperature-rise DMA (see Table 1).

## 4. Conclusions

UV-transparent, fast-cure encapsulants based on EVA and POE were examined regarding their chemical structure, formulation and crosslinking kinetics. By using FTIR spectrophotometry in transmission mode, a significantly lower amount of a polar comonomer, most likely based on acrylates, was ascertained for the investigated POE encapsulant. Moreover, the absorption peaks of the carbonate-based peroxide-crosslinking agents and the unsaturated C=C bonds of the co-curing agents and vinyl silanes were clearly discernable. The investigated EVA and POE films were based on slightly differing curing agent formulations. To describe the crosslinking behavior, dynamic and isothermal DMA experiments were carried out in torsional mode using a plate-plate rheometer. The temperature-rise experiment revealed about a 10 °C lower onset and gelation temperature for EVA. Hence, EVA is gelating earlier during the heating-up process of the lamination cycle. The consequence of gelation is a pronounced increase in the storage modulus and viscosity associated with reduced flow capability. At the offset of the curing, which was slightly above 160 °C, a factor of more than four higher storage modulus values were obtained for EVA.

Isothermal DMA experiments were conducted at temperatures ranging from 125 to 150 °C in 5 °C steps. EVA exhibited a more significant change in the storage modulus or complex viscosity compared to POE. Moreover, the gelation time at a defined temperature was a factor of four times longer for POE, which was therefore characterized by a retarded crosslinking behavior. These differences are presumably related to the higher co-monomer content and more tertiary carbon atoms in EVA. Nevertheless, a quite similar activation energy value of about 155 kJ/moles was obtained for both of the materials. Hence, the consideration of just the activation energy to describe the crosslinking kinetics of the encapsulants is not meaningful. By evaluating the time and temperature-dependent complex viscosity, the progress of the crosslinking values was deduced. The time to achieve a progress of crosslinking of 95%, which is recommended in photovoltaic module lamination, was up to a factor of 1.6 longer for POE (at 150 °C: 8.9 min for EVA vs. 14.6 min for POE).

The crosslinking kinetics study revealed a significant difference between UV-transparent, fast-cure EVA and POE. The provided data are of high relevance for the definition of photovoltaic module lamination parameters. In future research, focus will be given to the establishment of the correlations between critical lamination process parameters and the crosslinking kinetics data deduced on an encapsulation film level. Such a fundamental understanding would allow for an efficient and reliable adjustment of the lamination parameters based on the materials data. Finally, it is emphasized that dynamic mechanical analysis is a highly efficient characterization method for the quality assurance and shelf-life testing of encapsulant films.

## Figures and Tables

**Figure 1 polymers-14-01441-f001:**
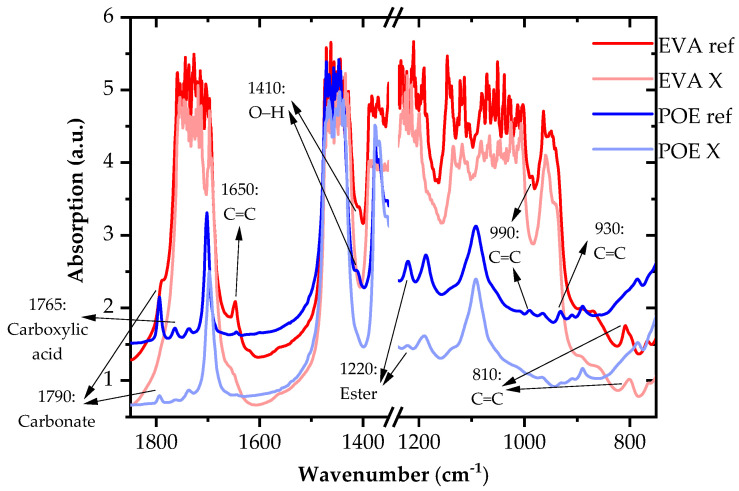
FTIR spectra of UV-transparent, fast-cure EVA and POE grades in the reference (ref) and fully crosslinked (X) state; decaying peaks or shoulders are numbered and highlighted with arrows.

**Figure 2 polymers-14-01441-f002:**
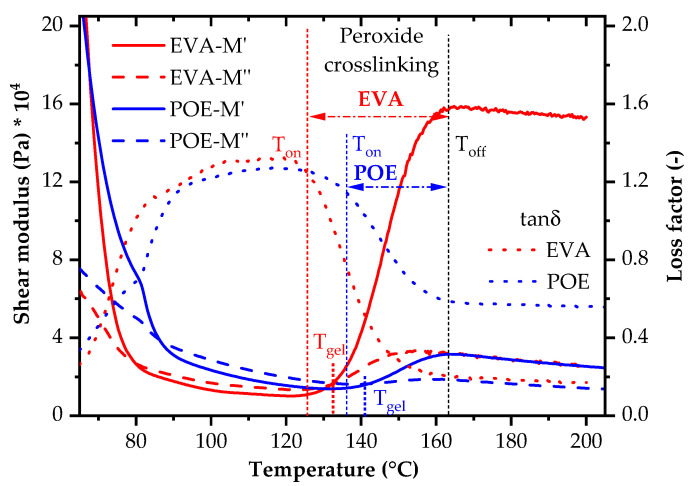
Temperature-rise DMA curves of the investigated peroxide-crosslinking EVA and POE grades (T_on_…onset temperature, T_gel_…gelation temperature, T_off_…offset temperature).

**Figure 3 polymers-14-01441-f003:**
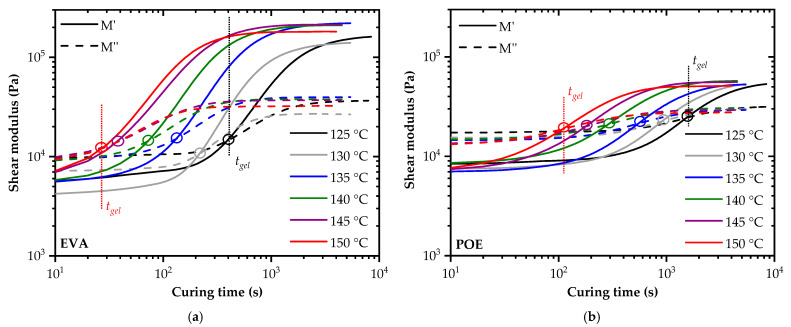
Storage and loss modulus curves of EVA (**a**) and POE (**b**) at different isothermal curing temperatures (*t_gel_*…gelation time).

**Figure 4 polymers-14-01441-f004:**
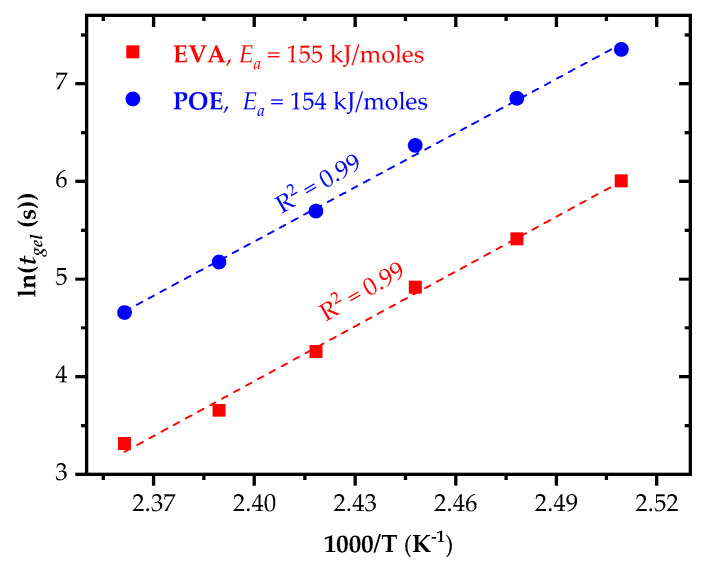
Arrhenius plots and activation energy for EVA and POE encapsulants.

**Figure 5 polymers-14-01441-f005:**
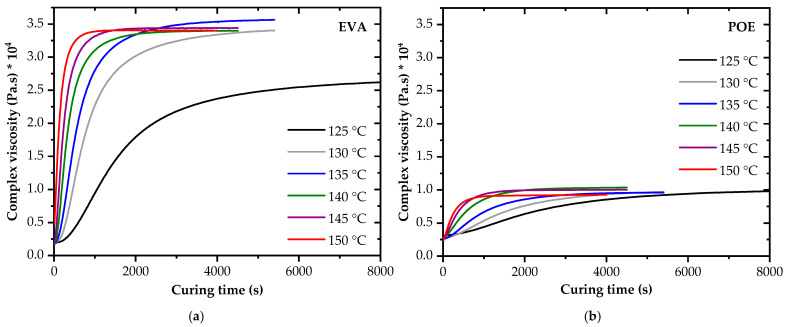
Complex viscosity of EVA (**a**) and POE (**b**) as a function of curing temperature and time.

**Figure 6 polymers-14-01441-f006:**
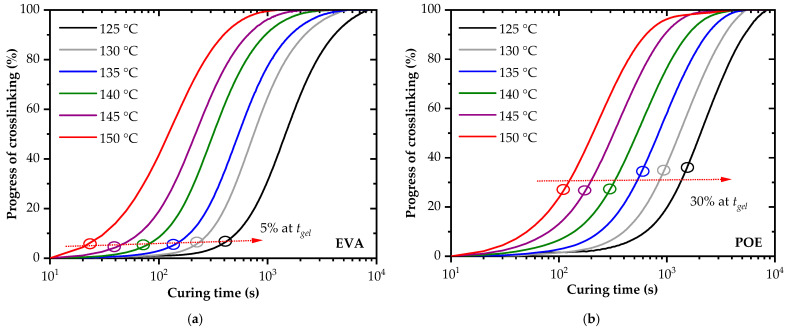
Progress of crosslinking of EVA (**a**) and POE (**b**) as a function of curing temperature and time (*t_gel_*…gelation time).

**Table 1 polymers-14-01441-t001:** Onset, gelation and offset temperatures as well as total curing time of UV-transparent, fast-cure EVA and POE encapsulants.

Encapsulant	T_on_, °C	T_gel_, °C	T_off_, °C	Cure Time, min
EVA	125	130	164	13
POE	135	140	162	9

**Table 2 polymers-14-01441-t002:** Temperature-dependent gelation time, progress of crosslinking at gelation time and cure time to achieve a progress of crosslinking of 95% for UV-transparent EVA and POE.

Temperature, °C	Gelation Time *t_gel_*, s		Progress of Crosslinking *X* at *t_gel_*, %		Cure Time for *X* = 95%, s	
	EVA	POE	EVA	POE	EVA	POE
125	405	1559	6	36	5370	6320
130	224	945	6	35	2995	4120
135	136	582	5	34	2275	2975
140	71	297	5	29	1325	2040
145	39	177	5	26	900	1240
150	28	111	7	25	535	875

## Data Availability

Not applicable.

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
