# Peer review of "Comparison of Crosslinking Kinetics of UV-Transparent Ethylene-Vinyl Acetate Copolymer and Polyolefin Elastomer Encapsulants"

_polymers, 2022, doi:10.3390/polym14071441_

Round 1

Reviewer 1 Report

In this manuscript, the authors studied the crosslinking kinetics of EVA and POE using DMA measurements. The manuscript is well written. I have some minor suggestions:

  1. Figure 1, I suggest the authors to label the selected peaks with frequencies for readers to better understand the structures of the films.
  2. Figure 2, it's very hard to clearly see the M" curves, same for Figure 3. 
  3. I suggest the authors to also include the tan delta curves in Figure 2. 
  4. Figure 4, I suggest the authors to use 1000/T as the x axis. 
  5. Some of the relevant references on EVA can be cited such as: Renewable and Sustainable Energy Reviews 81 (2018): 2299-2317; Polymer degradation and stability 95.5 (2010): 725-732.;  Journal of Macromolecular Science, Part B 54.12 (2015): 1515-1531..

Author Response

Dear Reviewer,

thanks a lot for providing the highly esteemed feedback to our research paper. All your recommendations were considered and the paper was revised accordingly. A rather good idea was to show also the tan delta curves. They were confirming the differences in crosslinking kinetics.

Best regards,

Gernot Wallner

Reviewer 2 Report

Work titled "Comparison of crosslinking kinetics of UV transparent ethylene vinyl acetate copolymer and polyolefin elastomer encapsulants" is very interesting and well prepared. All presented results are well presented, described and discussed with the literature.

My suggestions:

  • The general chemical structure of polymers should be presented
  • Line 105: ... an aliphatic (-C8H16-) central group
  • In my opinion, FTIR spectra of cross-linked systems should be added and discussed (to confirm proper cross-linking).

Author Response

Dear Reviewer,

thanks a lot for providing the highly esteemed feedback to our research paper. Nearly all your recommendations were considered and the paper was revised accordingly. A rather good idea was add the FTIR spectra of the crosslinked systems. By evaluation of several decreasing peaks valuable information as to the curing agent formulation could be deduced.

We decided to omit the presentation of the general chemical structure of the investigated polymers. I kindly ask for your understanding. For EVA, it's well described in the cited literature. For POE, the primary structure is still unknown. Based on the IR spectra it was not possible to clarify unambiguously which comonomers were polymerized (e.g., hexene/octene or methyl/ethyl or buthyl-acrylate). Presumably, it was even a termopolymer.

Best regards,

Gernot Wallner